# Core-Shell Graphitic Carbon Nitride/Zinc Phytate as a Novel Efficient Flame Retardant for Fire Safety and Smoke Suppression in Epoxy Resin

**DOI:** 10.3390/polym12010212

**Published:** 2020-01-15

**Authors:** Weiwei Zhang, Weihong Wu, Weihua Meng, Weiya Xie, Yumeng Cui, Jianzhong Xu, Hongqiang Qu

**Affiliations:** 1College of Chemistry and Environmental Science, Hebei University, The Flame Retardant Materials and Processing Technology Engineering Technology Research Center, Baoding 071002, China; zhangwei950130@163.com (W.Z.); mengweihua121@163.com (W.M.); xwy990123@163.com (W.X.); cuiyumenghbu@163.com (Y.C.); 2College of Science, Hebei Agricultural University, Baoding 071000, China; weiweibigq@163.com

**Keywords:** graphitic carbon nitride, zinc phytate, epoxy resin, flame retardancy

## Abstract

Novel core-shell graphitic carbon nitride/zinc phytate (g-C_3_N_4_/PAZn) flame retardant was simple synthetized using two-dimensional g-C_3_N_4_ and bio-based PAZn by self-assembly and incorporated into epoxy resin (EP) for improving the fire safety. The flame retardance and smoke suppression were investigated by cone calorimetry. The results indicated that g-C_3_N_4_/PAZn-EP displayed outstanding flame retardancy and smoke suppression, for example, the peak heat release rate and peak smoke production rate decreased by 71.38% and 25%, respectively. Furthermore, the flame retardancy mechanism was further explored by char residue and thermal stability analysis. It can be predicted that g-C_3_N_4_/PAZn will provide valuable reference about bio-based flame retardant.

## 1. Introduction

Epoxy resin (EP) is one of the most important resins because of its excellent moldability, corrosion resistance, adhesion, mechanical and electrical properties [1,2]. It is widely used in communications, automotive industry, semiconductor device, construction and other fields [3]. However, EP consists of flammable hydrocarbon chains, resulting in thermal and toxic hazards to life health and environment during combustion [4]. Therefore, it is urgent to develop a high-efficiency, environmentally friendly flame retardant to reduce the fire risk and broaden the application of EP composites.

Recently, plenty of experiments indicate that two-dimensional nanosheets materials have an advantage of effectively block the transmit of heat and smoke release due to their unique barrier effects, such as layered double hydroxide [5], graphene oxide [6] and graphitic carbon nitride (g-C_3_N_4_) [7]. The nanosheets structure plays a role of a physical barrier with a forming tortuous path between each layer, which can effectively prolong the conductivity path of heat and smoke release between the matrix and external environment [8,9]. Among them, g-C_3_N_4_ could decompose into nitrogen-containing non-flammable gas in fire, meanwhile it has low price, excellent thermal and chemical properties [10]. However, g-C_3_N_4_ is not enough to achieve the satisfactory effects when used alone as a flame retardant. Therefore, it is particularly important to further enhance the flame retardancy by hybridizing the g-C_3_N_4_ according to the principle of flame retardancy. Shi et al. found that g-C_3_N_4_ hybridized with organic aluminum hypophosphite could reduce significantly heat and smoke release rate of polystyrene, resulting in reducing the fire hazards [11]. Moreover, Shi et al. also reported that g-C_3_N_4_ combined with sodium alginate could promote the formation of stable char layer and enhance thermal stability and flame retardancy of composites [12].

Currently, bio-based flame retardant has received widespread attention due to the trend of environmentally friendly society. Additionally, phosphorus–nitrogen flame retardant could synergistically improve the flame retardancy [13]. Bio-based phytic acid (PA) composed by six phosphate groups was chelated with metal ions easily and provided acid as well as carbon sources [14]. Phosphorus-containing compounds are thermally decomposed to PO·, which can block combustion though quenching matrix combustion produce H· and HO·; on the other hand, phosphorus-containing compounds can catalyze the dehydration and carbonization reaction containing O–H compounds [15]. Meanwhile, the metal ions can also catalyze the formation of a stable cross-linked char layer effectively, preventing the release of smoke and further pyrolysis of polymers [16]. In fact, g-C_3_N_4_ and PAZn containing N, P and Zn flame retardant elements could be an ideal combination improving the flame retardancy of matrix.

Until now, there has been no research about g-C_3_N_4_/PAZn for improving flame retardant of EP. In this paper, efficient novel core-shell g-C_3_N_4_/PAZn was simple synthesized by calcination and chemical precipitation methods successfully and cured into EP to elevate the fire stately. The corresponding flame retardancy mechanism was proposed through thermogravimetric and residue char analysis. g-C_3_N_4_/PAZn will provide valuable reference about bio-based flame retardant and expand the application range of EP.

## 2. Materials and Methods

### 2.1. Materials

Melamine (99.5%) and methanol (99.7%) were purchased from Tianjin Kemiou Chemical Reagent Co. Ltd (Tianjin, China). PA (70.0%), Zinc nitrate hexahydrate (98.0%) and m-phenylenediamine (99.0%) were provided by Shanghai Aladdin biological technology Co., Ltd (Shanghai, China). Non-solvent EP-44 (SINOPEC, epoxide equivalent is 0.40–0.47) was obtained from Baling Petrochemical Corporation Branch, China Petrochemical Co., Ltd (Yueyang, China). 

### 2.2. Synthesis of g-C_3_N_4_/PAZn

g-C_3_N_4_/PAZn was synthesized by calcination and chemical precipitation methods. Firstly, melamine was taken in a porcelain boat and heated to 600 °C for 4 h at a heating rate of 5 °C/min to obtain light yellow g-C_3_N_4_ nanosheets. Then, 0.40 g of g-C_3_N_4_ nanosheets was dispersed in 60 mL methanol after stirred and sonicated for 10 min at 80 °C. 2.86 g PA and 7.72 g zinc nitrate hexahydrate were incorporated into 60 mL methanol and added dropwise into above solution, respectively. White precipitate g-C_3_N_4_/PAZn was generated after being stirred for 4 h at 80 °C. Finally, g-C_3_N_4_/PAZn was collected by washed with deionized water for three times and dried at 80 °C for 12 h. The formation mechanism of g-C_3_N_4_/PAZn is shown in Scheme 1. The NH_2_ functional group in g-C_3_N_4_ can react with OH in PA, PA can chelate Zn^2+^ to obtain g-C_3_N_4_/PAZn.

### 2.3. Preparation of EP Composites

EP composites were prepared according to a typical preparation method. 5 phr of synthetic flame retardants (g-C_3_N_4_, g-C_3_N_4_/PAZn) were added to pure EP slowly and stirred for 40 min at 60 °C. Generally speaking, the required mass of m-phenylenediamine = (relative molecular mass of m-phenylenediamine/number of active hydrogen) × epoxide equivalent. 11 phr m-phenylenediamine was added into EP composites and stirred for 20 min. Finally, EP composites was poured into the Teflon mold as fast as possible following curing under 80 °C for 2 h and 150 °C for 3.5 h to obtain g-C_3_N_4_-EP and g-C_3_N_4_/PAZn-EP, respectively. The composition of EP composites is shown in Table 1.

### 2.4. Characterization

The morphology of samples was analyzed by scanning electron microscope (SEM, JSM-7500F, JEOL, Tokyo, Japan) and transmission electron microscope (TEM, G2 F20, S-TWIN, Hillsboro, OR, USA). The crystal structure was determined by X-ray diffraction (XRD, D8-ADVANCE, Bruker, Karlsruhe, Germany) with the scanning from 10 to 90° at the speed of 10°/min. The functional groups were investigated by Fourier transform infrared spectra (FTIR, TENSOR 27, Bruker, Germany) spectra with the scanning wavelength range from 4000 to 400 cm^−1^. Thermostability of samples were carried out using thermogravimetric analysis (TGA, STA449C, Netzsch, Germany) from 50 to 800 °C at a heating rate of 10 °C/min. The limiting oxygen index (LOI) of samples was measured according to ASTM D2863 standard by general model JF-3 limiting oxygen index (Jiangning Analytical Instrument Company, Nanjing, China). The combustion behavior of composites was tested by cone calorimeter test (CCT, iCONE plus, Fire Testing Technology, West Sussex, UK) according to the ISO5660-1 standard under the instrument radiant power of 50 kW/m^2^. The structure of char residue was obtained by Raman spectrometer (Raman, XploRA Via-Reflex, HORIBA Jobin-Yvon Ltd, Paris, France) with an excitation wavelength of 514 nm. The tensile strength was performed by UTM4204 electronic universal testing machine (SUNS, Shenzhen, China), carrying out speed at 50 mm/min and scale distance was 25 mm. The impact test was tested by ZBC2000-B pendulum impact tester (the Winters Industrial Systems Co. Ltd., Shanghai, China), according to the ISO 180:2000. Differential scanning calorimetry (DSC) was performed by the Perkin Elmer Diamond DSC and the heating rate was 10 °C/min under N_2_ atmosphere.

## 3. Results and Discussion

### 3.1. Characterization of g-C_3_N_4_/PAZn

The XRD and FTIR were analyzed to determine the crystal structures and surface functional groups of materials. As shown in Figure 1a, two characteristic diffraction peaks located at 13.1° and 27.6° reflected the successful preparation of g-C_3_N_4_ [17]. The curve of g-C_3_N_4_/PAZn was similar to that of g-C_3_N_4_, while a small peak at 17° was ascribed to phytic acid. It′s worth noting that g-C_3_N_4_/PAZn appeared a lower and broader diffraction peak about 30°, demonstrating that the formation of amorphous PAZn [18]. The change of surface functional groups could further confirm the synthesis of g-C_3_N_4_/PAZn. As shown in Figure 1b, the broad absorption bands at 3470 and 3330 cm^−1^ were assigned to the stretching vibration of N–H and the absorption peak at 800 cm^−1^ were ascribed to triazine ring [19]. In comparison, these peaks seem to be disappeared, indicating the formation of –NH_3_^+^O and some new absorption peaks appeared in g-C_3_N_4_/PAZn [14]. The absorption peaks at 1404, 1253 and 1146 cm^−1^ were associated to the stretching vibration of C–O, P=O and P–O bonds, respectively [20]. Moreover, the characteristic bands of Zn salt were detected at 550 cm^−1^ [21]. This result could be verified by SEM and TEM analysis.

The morphology of g-C_3_N_4_ and g-C_3_N_4_/PAZn was characterized by SEM and TEM, as depicted in Figure 2. g-C_3_N_4_ is two-dimensional nanosheets structure, containing C and N elements. Compared with g-C_3_N_4_, the surface of g-C_3_N_4_/PAZn has an obvious coating layer, containing C, N, O, P and Zn elements, which is further confirmed that g-C_3_N_4_ was successfully coated by PAZn.

The thermal degradation behavior was related to flame retardancy and estimated by TGA and shown in Figure 3. g-C_3_N_4_ underwent a one-stage decomposition attributing to the decomposition of macromolecular chain and the char residues is only 1.22%. *T*_5%_ is defined as the initial decomposition temperature (which is the temperature corresponding weight loss of 5%). *T*_5%_ of g-C_3_N_4_ occurred at 650.8 °C, indicating that it has high thermal stability and play an effect in physical isolation. Comparing with pure g-C_3_N_4_, *T*_5%_ of g-C_3_N_4_/PAZn was obviously ahead and the char residues of g-C_3_N_4_/PAZn was improved to 43.42%. g-C_3_N_4_/PAZn underwent a two-stage decomposition, the first decomposition step between 250 and 350 °C, mainly because the phosphonate group was dehydrated and condensed into polyphosphoric compounds [22]. In the second decomposition step at between 550 and 650 °C it corresponds to the further degradation of polyphosphoric compounds into phosphorus-containing oxides and the degradation of g-C_3_N_4_ [23].

### 3.2. Fire and Smoke Hazards of EP Composites

The LOI is an effectively measure for evaluating the flame retardancy of EP composites, and the number of samples is 6. As shown in Figure 4, the LOI value of pure EP was 24.5%, indicating that pure EP is flammability. Obviously, the addition of flame retardant can improve the flame retardancy of EP composites effectively. The LOI value for g-C_3_N_4_-EP increased to 27.4% from 24.5% and g-C_3_N_4_/PAZn-EP further moved up to 28.3%, showing that g-C_3_N_4_/PAZn can reduce fire hazards and expand the application of EP [24]. 

CCT can provide relevant information about the flame retardancy and smoke suppression properties of EP composites in real fire situation. The heat release rate (HRR) and total heat release (THR) are an important index for assessing the fire hazards, as shown in Figure 5a,b and Table 2. HRR curves show that: There were two obvious peaks in EP, the first peak corresponds to the decomposition of EP chain; the second peak corresponds to the further decomposition of char [25]. The peak heat release rate (PHRR) values of g-C_3_N_4_-EP and g-C_3_N_4_/PAZn-EP decreased from 1458.14 kW/m^2^ (pure EP) to 906.28 and 417.26 kW/m^2^, which were 37.85% and 71.38% lower than pure EP, respectively. The THR of g-C_3_N_4_-EP and g-C_3_N_4_/PAZn-EP were 101.25 and 46.15 MJ/m^2^, which were 7.78% and 58.0% lower than pure EP (109.79 MJ/m^2^), respectively. Meanwhile, the residual char amount (R-mass) was improved obviously to 7.60% (g-C_3_N_4_-EP) and 12.90% (g-C_3_N_4_/PAZn-EP) compared with that of pure EP (4.58%). In addition, the time of ignition (TTI) and fire growth index (FGI) are important factors for estimating flame retardancy, prolonger TTI and lower FGI means higher fire safety. With the addition of g-C_3_N_4_/PAZn, lower HRR, THR, FGI and higher TTI, R-mass attesting that the flame retardancy of g-C_3_N_4_/PAZn-EP was improved [26]. This is due to the barrier and labyrinth effects of g-C_3_N_4_ nanosheets, which can limit heat transfer [27]. Furthermore, phosphorous and Zn^2+^ exhibited catalytic charring behavior to form a stable dense char layer and protect the matrix in the combustion process [28].

The smoke generation regards an important factor to human survival in fire. As shown in Figure 5c,d and Table 2, the smoke production rate (SPR) and peak smoke production rate (PSPR) of g-C_3_N_4_-EP were lower than that of pure EP, however the total smoke production (TSP) was higher than that of pure EP, which is probably due to the barrier and labyrinth effect of g-C_3_N_4_ nanosheets, meanwhile it can release containing nitrogen noncombustible gases at high temperature [29]. It is worth noting that the decomposition of g-C_3_N_4_/PAZn-EP was earlier than pure EP, showing PAZn exist excellent catalytic carbonization property at the early stage [30]. Furthermore, the barrier performance of char layer is beneficial to inhibit smoke and toxic gases, the TSP of g-C_3_N_4_/PAZn-EP was lower than g-C_3_N_4_-EP. Besides, the effective heat of combustion (EHC) indicates the level of burning of flammable gases in fire and lower EHC means lower combustible degree of gas [31]. The EHC decreased from 24.88 (pure EP) to 11.39 MJ kg^−1^ (g-C_3_N_4_/PAZn-EP), indicating the lower risk of fire hazard.

### 3.3. Research on Flame-Retardant Mechanism

#### 3.3.1. Thermal Stability Analysis of EP Composites

The thermal stability of EP composites under N_2_ is vital to assess flame retardant mechanism. TGA and derivative thermogravimetry analysis (DTG) curve and related data are shown in Figure 6 and Table 3, pure EP had one main degradation stage attributing to the decomposition of the C=C bond, and the maximum weight loss rate (*V*_max_) reached to 15.87%/min at 371.7 °C. The degradation stage of g-C_3_N_4_-EP and g-C_3_N_4_/PAZn-EP are similar to pure EP, while they have the lower *V*_max_ than that of pure EP. This is due to the barrier effect of g-C_3_N_4_. It is worth noting that g-C_3_N_4_-EP and g-C_3_N_4_/PAZn-EP have another degradation stage around at 600 °C, corresponds to the degradation of g-C_3_N_4_ [32]. Compared to pure EP and g-C_3_N_4_-EP, g-C_3_N_4_/PAZn-EP has the lowest *T*_5%_, the temperature of maximum decomposition (*T*_max_), *V*_max_ and highest R-mass. The results demonstrated that PAZn can catalyze the degradation and carbonization of EP form stable char layer at the early stage, which could protect the matrix to degrade and prevent the transfer of oxygen, combustible gas and external heat thus improving the thermal stability of EP [33].

#### 3.3.2. Char Residue Analysis 

To further explore the flame retardancy mechanism, the char residues were examined by digital photograph and SEM. As shown in Figure 7a–c, high quality and integrity char layer could effectively prevent transmit of heat and smoke [34]. The results showed that the char of pure EP had a loose and fragmentary structure with many holes and cracks on surface because of the rapid volatilization during intense combustion. With the addition of g-C_3_N_4_, the quality of char residue improved slightly with many holes, which was attributed to the pyrolytic decomposition of g-C_3_N_4_. g-C_3_N_4_/PAZn-EP displayed a continuous and density char, indicated that it had good catalytic performance for char residues, which could provide a barrier to protect matrix, exhibiting excellent flame retardant and smoke suppression behavior [35]. The results are consistent with the CCT results. In addition, the char residues of EP composites were further analyzed by Raman spectra (Figure 7a′–c′). The Raman spectra exhibited two protruding peaks at 1350 and 1581 cm^−1^, which are defined to the D band and G band of graphite, respectively. Generally, the area ratio (*I*_D_/*I*_G_) of D to G band can be judged the graphitization degree of the char residue [36]. The value of *I*_D_/*I*_G_ of pure EP was 2.70, whereas the values for g-C_3_N_4_-EP and g-C_3_N_4_/PAZn-EP were 3.20 and 3.39, respectively. The higher *I*_D_/*I*_G_ value indicates that the more lattice defects and the smaller size of char residues microstructures, which can protect the EP during combustion effectively [14].

### 3.4. Mechanical Property

From the research of LOI and CCT, it was known that flame retardancy was enhanced greatly by adding flame retardant. However, the mechanical property was a vital factor that must be considered in practical application. To research the mechanical property influence of flame retardant, the tensile and impact tests of EP composites were shown in Table 4. The tensile strength, elongation at break and impact strength of pure EP were 57.70 MPa, 14.67% and 50.83 kJ/m^2^, respectively. Due to g-C_3_N_4_ and g-C_3_N_4_/PAZn are additive flame retardant, had poor adhesion strength with EP, the tensile strength, elongation at break and impact strength of g-C_3_N_4_-EP and g-C_3_N_4_/PAZn-EP are reduced to some extent. With the addition of g-C_3_N_4_/PAZn, the tensile strength, elongation at break and impact strength of g-C_3_N_4_/PAZn-EP are 35.11 MPa, 6.49% and 26.04 kJ/m^2^, which indicates that g-C_3_N_4_/PAZn can cause stress concentration and block the molecular chains movement [37]. However, it still has a certain intensity. In comparison, g-C_3_N_4_ was added to EP, the corresponding values did not reduce significantly, this is because of amino groups on the surface of g-C_3_N_4_ participate in the reaction in EP curing process [38]. The glass-transition temperature (*T*_g_) was related to the mechanical property of EP composites and tested by DSC, as shown in Figure 8. The *T*_g_ of pure EP was 152.3 °C, and *T*_g_ value decreased with the addition of flame retardants, indicating that flame retardants can affect the micro-Brownian motion of EP molecular chains. The separate *T*_g_ of all the composites confirmed that the composites were a homogeneous phase. The *T*_g_ of g-C_3_N_4_-EP (151.7 °C) was higher than that of g-C_3_N_4_/PAZn-EP (149.4 °C), which ascribed to the strong interfacial interaction and the enhanced crosslink density both from the reaction between amino groups (g-C_3_N_4_) and EP [39].

## 4. Conclusions

In this paper, an efficient novel flame retardant core-shell g-C_3_N_4_/PAZn was successfully synthesized by simple calcination and chemical precipitation method. The CCT results proved that g-C_3_N_4_/PAZn displays outstanding flame retardancy and smoke suppression properties, the values of PHRR and PSPR decreased by 71.38% and 25%, respectively. On one hand, g-C_3_N_4_ play physical barrier role to protect heat and smoke form spreading, meanwhile it releases non-combustible gases for reducing the concentration of oxygen and combustible gases. On the other hand, PAZn can catalyze the degradation and carbonization of EP form stable char layer, which promotes the formation of dense char layer to protect EP during combustion effectively.

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
