# Peer review of "Core-Shell Graphitic Carbon Nitride/Zinc Phytate as a Novel Efficient Flame Retardant for Fire Safety and Smoke Suppression in Epoxy Resin"

_polymers, 2020, doi:10.3390/polym12010212_

Round 1

Reviewer 1 Report

The authors have performed the changes I suggested. In my opinion the article is now valuable to be published and much more clear than before.

Author Response

We sincerely thank you very much for your kind help and recognition of our manuscript work.

Reviewer 2 Report

1.-Mention in the abstract that characterizations of X-ray diffraction (XRD), fourier transform infrared spectra (FTIR), scanning electron microscopy (SEM) and transmission electron microscope (TEM) and limiting oxygen index (LOI) and cone were made calorimeter test (CCT). Because they are not mentioned and it seems that there was no complete study when reading the abstract

2.- Figure 1. (a) XRD patterns of g-C3N4 and g-C3N4 / PAZn; (b) FTIR spectra of g-C3N4 and g-C3N4 / PAZn
check in which units the transmittance is reported and accommodate the caption

3.- Check the Y axis units, which must be reported as weight (%).

4.-Check the accommodation of the manuscript, that the figure feet are not cut.

Author Response

We sincerely thank you very much for your valuable advice and strict scientific attitude of our manuscript. We have tried our best to revise our manuscript and some explanations were attached in this letter. The amendments we have made were marked as those parts in red colour in this revised manuscript.

This manuscript is a resubmission of an earlier submission. The following is a list of the peer review reports and author responses from that submission.

Round 1

Reviewer 1 Report

1.- Mention in the abstract that characterizations of X-ray diffraction (XRD), fourier transform infrared spectra (FTIR), scanning electron microscopy (SEM) and transmission electron microscope (TEM) and limiting oxygen index (LOI) and cone were made calorimeter test (CCT). Because they are not mentioned and it seems that there was no complete study when reading the abstract.
. 2,. In Scheme 1. The synthesis process of g-C3N4 / PAZn / ZIF-8. Chemical structures and bonds or electrons are not properly appreciated, the quality of this figure could be greatly improved.
3.- The Figure 2. SEM images of a g-C3N4, b g-C3N4 / PAZn, c g-C3N4 / PAZn / ZIF-8; TEM images of a ’g-C3N4, b’ g-C3N4 / PAZn, c ’g-C3N4 / PAZn / ZIF-8; elementary mapping of d ’g-C3N4 / PAZn / ZIF-8. It is of very poor quality, the SEM images are not clearly visible nor the increases to which they are taken and the elementary analysis graph is very blurred and the elements present are not visible, besides the units of the X-axis are missing and Y.

4.- In the preparation of the compounds, the contents of each sample to be analyzed in a table could be properly specified, for a better understanding of the differences that each compound may have. Lines 108 and 109.

5.- The quality and clarity of Figure 1b) FTIR spectrum is poor and in some cases it is not possible to distinguish the links to which reference is being made and the identification of the samples. In this same figure, the units of the Y axis are missing, which in this case should be transmittance.

6.- in line 97 they are not separated 30 and min
7.- on line 90 they are not separated by 5°C, change by 5 °C

8.- Check the Y axis units Figure 3. TGA curves of g-C3N4, g-C3N4 / PAZn, g-C3N4 / PAZn / ZIF-8. Which must be reported as weight (%).
9.- Review all the cases in which the unit number and ªC are together, for example line 162, change 250°C-350°C to 250 ° C-350 ° C. Review all similar cases in the document, for example on line 201 and 202 that indicates 7.60% (g-C3N4-EP), 12.90% (g-C3N4 / PAZn-EP), and 11.10%.

10.- separate 43.42% by 43.42 %. Review all similar cases in the document
.
11.- A space is missing in temperature [36], line 210.

12.- The DOI of a large part of the references presented is missing.

Reviewer 2 Report

The article “Core-shell Graphitic Carbon Nitride/Zinc Phytate/Metal organic Framework Nanoparticles for Flame Retardant and Smoke Suppression Properties in Epoxy Resin” study the effect of nanoparticles prepared in the fire retardancy of epoxy composites. Although the topic is interesting and the results in fire retardancy have been demonstrated, the work should be deeply revised since the results they obtained are not clear. According to that, I recommend to reject the article.

The following points should be considered to improve the quality of the article:

The English language should be edited. There are many grammatical mistakes. There is a lack of information about the type of the epoxy resin used. Please, add the name of the resin, the brand and the epoxy equivalent in Materials section. It is for me quite strange that the epoxy resin seems to be cured by the addition of 11 phr of m-phenylenediamine. This curing agent is usually stoichiometric and therefore the amount of amine should be calculated on the basis of epoxy equivalent of resin. Please comment on it, and calculate the stoichiometry you are using. If the authors add 11 phr of amine, it is necessary to know if it includes the resin and flame retardant modifiers or only the resin. The same for the flame retardant agents. In case of flame retardant agents the phr added consider the resin or the mixture resin and amine all together. Please, comment on it. Figure 2. It is quite difficult to see the magnifications of the SEM micrographs, especially those of the second line. In the elemental mapping, the atoms in each peak cannot be seen clearly. In fact, I am not convinced of the nanometer size of the particle prepared. How many tests for each sample have the authors performed of LOI index? Which is the confidence level or the dispersion of the data obtained? Figure 4 shows the enhancement of LOI value on adding different fire retardant agents. In my opinion, there is no much difference between the values of 28.3 and 28.7 and therefore the conclusion about the reduction of fire hazards on adding g-C3N4/PAZn/ZIF-8 is not as clear. Please clarify this point. This is especially dubious by looking to the CCT values with no much difference between composites containing ZIF-8 or not. Figure 5 is too small and difficult to read. The discussion on the flame retardancy is too long and complex to be followed by most of the readers. Please summarize it and highlight the most important results. There is no information about the Tg of these composites. From the structure of the resin (unknow) and the degree of curing achieved (not know because we don’t know the stoichiometric defect or excess of amine) I have no idea if the material is rubbery or glassy. This is important when the mechanical properties are evaluated, to know how far of the Tg, the measurements were done. There are no captions in Figure 9. Please confirm to which of the samples corresponds a, b, c and d images. From the SEM images in Figure 9 it is clear that there is a really bad compatibility. Why the authors have not tried to reach a better dispersion by sonication? Although the authors qualify the flame retardant as Nanoparticles, finally they get a microcomposite with a bad dispersion. Although the fire retardancy improves (although not spectacularly) the tension-stress properties get worse. What about impact resistance or toughness measurements? I do not fully agree with the statement in the conclusions: “The CCT results proved that g-C3N4/PAZn/ZIF-8 display outstanding flame retardancy and smoke suppression properties for combining actions of physical barrier, the gas phase and the condensed phase mechanism”. If we compared the composites with g-C3N4/PAZn/ZIF-8 with those with g-C3N4/PAZn, we cannot see many differences and even some results of fire retardancy and thermal stability are just on the contrary sense. The improvements are only small and this conclusion becomes not true.